# Preliminary Biocompatibility Tests of Poly-ε-Caprolactone/Silver Nanofibers in Wistar Rats

**DOI:** 10.3390/polym13071135

**Published:** 2021-04-02

**Authors:** Oskar Álvarez-Ortega, Luis Roberto Ruiz-Ramírez, Jesús Alberto Garibay-Alvarado, Alejandro Donohue-Cornejo, León Francisco Espinosa-Cristóbal, Juan Carlos Cuevas-González, Simón Yobanny Reyes-López

**Affiliations:** 1Departamento de Ciencias Químico Biológicas, Instituto de Ciencias Biomédicas, Universidad Autónoma de Ciudad Juárez, 32315 Chih, Mexico; al204562@alumnos.uacj.mx (O.Á.-O.); al199127@alumnos.uacj.mx (L.R.R.-R.); al199073@alumnos.uacj.mx (J.A.G.-A.); 2Departamento de Estomatología, Instituto de Ciencias Biomédicas, Universidad Autónoma de Ciudad Juárez, 32315 Chih, Mexico; adonohue@uacj.mx (A.D.-C.); leon.espinosa@uacj.mx (L.F.E.-C.); juan.cuevas@uacj.mx (J.C.C.-G.)

**Keywords:** biocompatibility, PCL/Ag, inflammation, foreign body reaction

## Abstract

Currently, nanotechnology is perceived as a promising science that produces materials with diverse unique properties at a nanometric scale. Biocompatibility tests of poly-ε-caprolactone nanofibers, embedded with silver nanoparticles manufactured by means of the electrospinning technique, were carried out in Wistar rats to be used as oral dressings for the eradication of bacteria. Solutions of 12.5, 25, 50 and 100 mM of silver nitrate were made using *N*-dimethylformamide (DMF) and tetrahydrofuran (THF) as reducing solvents with 8% of poly-ε-caprolactone (PCL) polymer. The solutions were electrospun, and the nanofibers obtained in the process were characterized by infrared spectroscopy, Raman spectroscopy, dark field optical microscopy, scanning electron microscopy and X-ray scattering spectroscopy. The nanofibers had an average diameter of 400 ± 100 nm. Once the characterization of the material was done, three implants of each concentration of the nanofibers were formed and placed in the subcutaneous tissue of the rats. Three experimental subjects were used, leaving the material in them for a length of two, four and six weeks, respectively. The rats showed good healing, with the lesions completely healed at four weeks after implantation. After that time, biopsies were taken, and histopathological sections were made to evaluate the inflammatory infiltrate. The tissues of the rats presented chronic inflammatory infiltrate composed mainly of lymphocytes and giant multinucleated cells. The material was rejected by the rats when a layer of collagen and fibroblasts was produced, coating the material, a process characteristic of a foreign body reaction.

## 1. Introduction

Over the years, there has been a need to correct problems related to tissue loss due to trauma, disease or deterioration. This has oriented research to the development of materials that can be used to reproduce the function of living tissues in biological systems in a safe and physiologically acceptable way. These materials are known as biomaterials, products used to reproduce the function of living tissues in biological systems safely [1]. When a biomaterial comes into contact with living tissues, it should not produce any type of alteration in them. For this reason, a biomaterial must meet certain requirements that are encompassed by the concept of biocompatibility, including tolerance to said material, its short- and long-term biostability, the maintenance of its properties and the chemical and physical structure of the biological environment during the time that the biomaterial remains in the organism.

To ensure the functioning and success of a new biomaterial, it is necessary for the material to pass a series of tests that are carried out before clinical tests. These tests are mechanical simulations, in vitro tests, toxicological tests and in vivo tests in experimental animals. The set of these tests will determine the biocompatibility of the materials [2]. 3-(4,5-dimethylthiazol-2-yl)-2,5-diphenyltetrazolium bromide (MTT) is used as an indicator of overall cytotoxicity. This method is based on the ability of living cells to reduce dissolved MTT (yellow) into insoluble formazan (blue) in the presence of mitochondrial succinate dehydrogenase [3]. In vivo tests in experimental animals in subcutaneous tissue are preliminary biocompatibility tests, which provide valuable information on the reactions that a material can generate at a histological level. Considering the complexity of tissue reactions, the diversity of biomaterials and the great variety of their applications, it is said that the ideal biomaterial must have a dynamic surface that does not generate histological changes at the implant interface, changes that could happen if the implant were not present, for example, reabsorbing collagen sutures after they have performed their function, without causing histological alterations [4].

Currently, most postoperative oral infections are caused by endogenous bacteria from the normal flora of the body. The chosen bactericidal agent must be able to fight this infection without causing a harmful effect on the body. Increased use of antibiotics is the most important factor in the appearance of various classes of bacterial resistance [5]. The resistance of bacteria to antibiotics, antiseptics and disinfectants is a public health problem. Since the discovery of antibiotics, multiple bacteria have been able to resist their antimicrobial action. An example of this is the case of the bacterium *Staphylococcus aureus*, which was previously, in most of its strains, sensitive to penicillin, but today almost all its hospital strains show resistance to penicillin and its derivatives [6]. It is essential to create new strategies aimed at fighting bacterial infections due to the resistance generated in recent years.

Nanotechnology is currently perceived as a promising science offering new materials with different properties due to the scale at which it is found. Biomedical materials having metallic nanostructures with antimicrobial properties are presented as a promising strategy to combat these antibiotic-resistant microorganisms and have led to a great boom in tissue engineering in the pharmaceutical industry [7]. Among the mentioned nanostructures, we find that silver nanoparticles (AgNPs) have the ability to adhere to the bacterial cell membrane, altering its permeability and respiratory functions, eliminating them in the process [8]. AgNPs was shown to stop bacterial growth and contribute to tissue regeneration, thus avoiding limb amputation in dozens of patients [9]. According to recent research, silver nanoparticles are known to interact with the cell surface of bacteria. The ions released by the nanoparticles interact with the surface, causing structural changes, increasing the permeability of the membrane and leading to cell death [10]. These ions interact with the exposed sulfhydryl groups to stop bacterial replication. The cell cycle is stopped due to DNA damage, and bacteria die from oxidative stress [11].

In recent years, tissue engineering has generated antimicrobial scaffolds, which are used in bone regeneration and as dressings for wounds. The biomaterials used contain silver nanoparticles that have demonstrated that they have bactericidal activity against Gram-positive and -negative bacteria, in addition to having a minimal level of toxicity towards rat and human cells [12]. Currently, innovative antimicrobial agents are developed and improved using nanotechnology to control and mitigate resistant microorganisms, because metal nanoparticles or metal oxides are toxic to bacteria even at low concentrations [13]. The integration of silver nanoparticles in polymer matrices has improved its antimicrobial performance, resulting in the search for new composites with improved bactericidal properties [14]. In this project, polycaprolactone was used as a matrix and silver as an antimicrobial agent, because it has bactericidal activity. Therefore, the aim is to use the poly-ε-caprolactone–silver material as an oral dressing in oral operations for the eradication of infection-causing bacteria, replacing the use of antibiotics in addition to promoting the regeneration of tissue.

## 2. Materials and Methods

### 2.1. Preparation of Solutions

The composite was fabricated using a modified version of the methodology by Pazos-Ortiz et al. [15]. A total of 5 solutions measuring 10 mL with a 7:3 ratio of dimethylformamide (DMF, Sigma-Aldrich, St. Louis, MO, USA, ≥99.8%) and tetrahydrofuran (THF, Sigma-Aldrich, ≥99.5%) were prepared in amber bottles. The solutions underwent constant magnetic stirring for 10 minutes. Then amounts of silver nitrate (AgNO_3_, Sigma-Aldrich, ≥99%) were weighed, and the different masses were added to the 10 mL DMF/THF solutions, obtaining concentrations of 12.5, 25, 50 and 100 mM. 0.8 g of poly-ε-caprolactone (PCL) was added to the solutions and mixed for one hour at 50 °C. Once the solutions were prepared, the electrospinning process was continued [15]. Particle size and distribution were calculated by dynamic light scattering (DLS) in a nanoparticle analyzer, SZ-100 (HORIBA, Kyoto, Japan).

### 2.2. Electrospinning

PCL/Ag solutions were placed in a 10 mL volume container using a 14.5 mm diameter glass syringe. A 0.80 mm diameter stainless steel needle was attached to the syringe. The syringe was placed in the injector pump (Cole-Parmer, Vernon Hills, IL, USA), which was programmed with a flow in a range of 7 to 10 µL/min. An aluminum foil was placed in a rotating collector; the distance between the collector and the tip of the needle was 8 cm. Finally, the positive pole of a power source was placed on the needle and the negative pole on the rotary collector, and a voltage in the range of 10 to 14 kV was administered. The electrospinning process was carried out for an estimated time of 3–5 h to obtain the nanofibers. This process was done for each solution prepared (12.5, 25, 50 and 100 mM). After the process, the nanofibers were stored in previously labeled plastic for storage.

### 2.3. Nanofiber Characterization

The PCL/Ag fibers obtained by the electrospinning technique were characterized by infrared spectroscopy (FTIR) using an Alpha platinum-ATR Bruker spectrometer (Bruker Corporation, Billerica, MA, USA); a WITec’s Raman spectrophotometer Alpha-300 R (WITec GmbH, Ulm, Germany) with a 532 nm laser was used for Raman spectroscopy. For scanning electron microscopy (SEM), a JEOL JSM-6400 device (JEOL Ltd., Tokyo, Japan) operated at 20 kV equipped with an X-ray scattering spectrometer (EDX) was used. The magnifications used were 5000×, 10,000× and 20,000×. For each characterization technique, a sample of each fiber measuring 1 cm × 1 cm was used.

### 2.4. In Vivo Biocompatibility Test

Wistar rats (four- or three-month-old males), weighing approximately from 150 to 300 g, were used, divided into three groups, and one rat was used as a control. Each of the rats was housed individually in conditions established by the American Veterinary Medical Association’s (AVMA) and NOM-062-ZOO-1999 technical specifications for the production, care and use of laboratory animals during the entire process of the experimental phase.

Each of the rats was housed individually in conditions established by the Institutional Animal Care and Use Committee (IACUC), Comite Institucional de Etica y Bioetica, Universidad Autonoma de Ciudad Juarez (CIEB-UACJ) and the NOM-062-ZOO-1999 technical specifications for the production, care and use of laboratory animals during the entire process of the experimental phase (date of committee approval: 23.10.2017, project: CIBE-2017-2-84).

It was carefully verified that the rats were fully anesthetized and that the personnel performing the procedure were fully trained. Three fractions were taken from each concentration of the nanofibers. The implants measured approximately 5 mm in length by 1.3 mm in internal diameter. Once the implants were made, they were UV-sterilized for a half-hour to avoid any type of contamination during the implantation process.

### 2.5. Implant Placement

For the anesthetic process, an intramuscular injection of ketamine/xylazine (0.12/0.01 mL, PiSa Agropecuaria, Hidalgo, Mexico) was performed in one of the hind legs of the rats. After anesthetizing the rats, the dorsal area was shaved and then covered with iodine (Dynarex, Orangeburg, NY, USA, ≥99%) to avoid any infection during the surgery. After applying anesthesia, the material was implanted. With the help of a previously sterilized veterinary surgical kit, 4 incisions were made in the dorsal part of the rats. Once the PCL/Ag was implanted, the wounds were sutured with 2 points per incision.

### 2.6. Histological Samples

Euthanasia was done at 2, 4 and 6 weeks after implantation (one rat every 2 weeks). Intravenous injection of sodium pentobarbital is the preferred method for euthanizing horses, dogs, cats and rodents. A quick and painless death is caused in the body [16]. An overdose of sodium pentobarbital (PiSa Agropecuaria) was administered intraperitoneally to rats. Small incisions were made in the implantation areas. Four biopsies were taken, one from each concentration of PCL/Ag, and were deposited in formalin (Sigma Aldrich, ≥99%) for preservation. The criteria for euthanasia coincided with the experimental endpoints to reduce the pain or distress caused by the experimental condition. Once the animals’ tissues were obtained, they were placed in 10% formalin (Drotasa, Mexico, Mexico) to preserve and fix the tissue until its subsequent staining. The pathological changes observed in the animal for the effects of the treatment were carried out prior to the main experiment to allow better results and to minimize the animals’ pain while the study was completed. All veterinarians performing euthanasia needed to be properly trained. Individuals also needed to ensure that the animals were dead before their carcasses were disposed of.

For inclusion in paraffin, the biopsies were placed in cassettes to be subsequently dehydrated by ethanol (pure solution, 96%) with a gradation increasing to 100%. After doing so, the samples were transferred to xylene (Sigma-Aldrich, ≥99%), which was used as an intermediate reagent. The cassettes were taken and transferred to paraffin previously liquefied in an oven. Liquid paraffin was poured into various molds. Samples were introduced and refrigerated at 4 °C for hardening. The paraffin blocks with the tissues were cut and, once the cuts were obtained, they were placed in a 37 °C water bath to warm them up. Afterwards, the cuts were placed in an oven at 60 °C for the deparaffinization process. Tissues were rehydrated with decreasing ethanol until they reached water. Tissues were dipped in hematoxylin (Sigma Aldrich) for 3 minutes, washed with water, then dipped in eosin (Sigma Aldrich) for 30 sand washed with ethanol to remove traces of dyes. Tissues were dehydrated again with increasing ethanol. Once the coverslip was placed on the slide with the histopathological cut, the tissues were observed under a microscope. Finally, the plates were observed in an Oxion light microscope at 10× and 40× and photos were taken by a Cemex camera (Euromex, Chicago, IL, USA).

## 3. Results and Discussion

The four polymer solutions were spun by the electrospinning technique to obtain the nanocomposite. The fibers were made under specific conditions of voltage, distance between the injection source and the collecting plate, humidity, and flow rate. In the synthesis process of the PCL/Ag nanofibers, a polymeric solution reduced by DMF was used. DMF is a compound of organic origin used as a solvent and reducing agent in various processes in which silver salts are involved, in this case silver nitrate. Reduction with DMF is one of the most used and one of the best techniques for the reduction of Ag^+^, compared to other organic solvents such as ethanol [17]. The silver nanoparticles synthesized by this simple reduction method produced a slightly viscous reddish-brown solution that, when diluted, turned yellow. The particle size was determined by dynamic light scattering, around 6 ± 2.9 nm with zeta potential of −55 ± 3.4 mV, indicating that the silver nanoparticles have good stability.

The infrared spectra obtained for each of the nanofibers (Figure 1) identify the intensity of the characteristic functional groups of the polymer, the carbonyl group, the ether group and the ester. Similar intensities are observed in the spectra obtained for the PCL and PCL/Ag nanofibers. Multiple stretch bands can be observed between 1000 and 1500 cm^−1^; a band with great intensity can be observed around 1700 cm^−1^, and bands with less intensity can be observed between 2700 and 3000 cm^−1^. In the infrared spectrum, stretch bands of the carbonyl group (C=O), characteristic of the PCL, are observed between 1700 and 730 cm^−1^. The bands that appear in the region between 2800 and 3000 cm^−1^ are related to carbon/hydrogen (C–H) bonds. Bands do not appear between the 2500 and 2000 cm^−1^ region; this region is characterized by the absence of bands [18]. In the 100 mM PCL/Ag spectrum, there is a small stretch in this region with weak intensity; Mondragón Cortez affirms that the bands that appear in this region are not relevant in the interpretation of the spectrum. Bands of multiple types of bond vibrations appear in the 1500 to 600 cm^−1^ regions, making it difficult to assign an origin to particular bands. The band at 1294 cm^−1^ corresponds to the ether group (COC), and the band at 1190 cm^−1^ to the ester group (OCO). Between 1280 and 1295 cm^−1^ are the stretch bands for single bonds such as alkanes (CC) and carbonyls (CO) [19]. In the IR spectrum of PCL/Ag samples, no differences are observed, thus indicating no chemical modification of the structure of PCL, therefore, a physical union in the PCL/Ag follows.

Raman spectroscopy is a versatile method that allows us to study the degree of amorphous and crystalline phases of nanocomposite membranes. The Raman spectrum of PCL/Ag nanofibers is shown in Figure 2 [20]. The intense band at 218 cm^−1^ is generally assigned to the silver–oxygen compound (Ag–O) or to the ionic species absorbed in the polymer. Several narrow peaks at 934 cm^−1^ correspond to the ester group (*v*C–COO) and others within the 1000–1150 cm^−1^ spectral range correspond to the stretching of the skeleton. The peaks at 1280–1350 cm^−1^ (ωCH_2_), 1400–1500 cm^−1^ (δCH2) and 2800–3000 cm^−1^ (*v*CH) refer to the crystalline fraction of the PCL. The wide peak at 825 cm^−1^ indicates that the amorphous phase is present in the PCL fibers [21]. According to the results of infrared, it is found that the formed composite is a mixture of silver particles and polymer without any change in the structure of the PCL.

The PCL/Ag nanofibers presented a smooth, cylindrical surface, without the presence of pores, with random orientation, without the presence of fractures and branched, as can be seen in Figure 3. The branching occurs due to smaller secondary jets formed in the path of the solution to the collecting plate thanks to alterations in the voltage and in the elongation of the solution [22]. The diameter of the nanofibers decreased with increasing silver concentration. This is attributed to an increase in the electrical charge and conductivity of the solution, resulting in a smaller diameter in the fibers. Its formation is controlled by voltage because it breaks the surface tension of the solution [14]. At each concentration, 500 fibers were measured, presenting an average diameter of 492 nm for the 12.5 mM PCL/Ag fiber, 449 nm for the 25 mM fiber, 530 nm for the 50 mM fiber and 504 nm for the 100 mM fiber. The EDX spectrum of the PCL nanofiber shown in Figure 3c shows a peak between 0.1 and 0.4 keV for carbon and between 0.5 and 0.6 keV for oxygen. Another peak can be seen in the range of 1.5 KeV belonging to aluminum, typical of the collecting plate where the electrospinning process was carried out. Peaks between 0.1 and 0.4 keV for carbon and between 0.5 and 0.6 keV for oxygen, characteristic of PCL, are shown in all EDX spectra of PCL/Ag nanofibers. In addition, other peaks in the 3 keV range, belonging to silver, are seen [23].

Applying silver to experimentally induced wounds in rats shows rapid and efficient healing. If silver has a beneficial effect on the healing process, it can serve as a collaborator in the treatment of surgical wounds. The agrees with the results obtained from healing since, 2 weeks after implantation, good healing was already observed, and at 4 weeks after implantation, the lesions were completely healed, and at 6 weeks, there were no traces of lesions, as can be seen in Figure 4.

In Figure 5, the PCL/Ag implants were not completely degraded and were encapsulated by a layer of cells of the immune system. The degradation time of the implants must be less than the cell growth time. It is assumed that the implanted material must have totally degraded by the time the lesion healed, which suggests that the organism rejected the nanofiber implant. This body response to PCL/Ag is due to the shape of the implants made. The implants were cylindrical in shape with several layers of thickness. The response to the material could have been different if a simple implant had been implanted instead of a massive one. Furthermore, the reaction that the tissue has depends on the amount that is implanted [1].

Biodegradable polymers of natural origin are ideal for carrying out tissue engineering processes due to their low immunogenic potential and their ability to interact with tissue. On the other hand, PCL is a synthetic polymer with long degradation times depending on its molecular weight. PCL can be biodegraded by living organisms in the open air such as bacteria and fungi, but it cannot be biodegraded in animals or human bodies due to a lack of adequate enzymes to carry out the process. It is not ruled out that it cannot be degraded in the body, but rather that the process takes longer, spreading mainly by hydrolytic means [24]. The morphology of the material influences the process of its degradation. Hydrolysis-prone functional groups must be accessible to polymer-colonizing enzymes, bacteria or fungi. High molecular weight polymers are less likely to be degraded [25].

When lesions were made in the rat tissue, a natural inflammatory response was generated due to the loss of tissue continuity, caused by lesions in the vasculature found in the subcutaneous connective tissues [26]. Inflammation occurs as an early response against foreign agents, relying on the production of proinflammatory cytokines (IL-1, IL-2 and IL-6), on the activation of the immune system and on the release of prostaglandins and chemotactic substances. Researchers recorded anti-inflammatory activity in rats using silver, showing rapid healing and an improved aesthetic appearance produced in a dose-dependent manner. This agrees with the scarring presented in the rats since, at four weeks after implantation, the lesions presented efficient healing. The physical restoration of the internal and external structures of the organism is involved in the healing process. In this process, cellular interactions take place, an inflammatory response occurs, there is participation of fibroblasts, etc. When an appropriate treatment is applied, such as the use of silver, the healing process can be accelerated, managing to prevent infections, abscess formation and even necrosis [27].

Figure 6 shows the tissues six weeks after implantation, when a foreign body reaction is observed with an intense chronic type of inflammatory infiltrate consisting mainly of lymphocytes, multinucleated giant cells and the formation of blood vessels (angiogenesis). The response generated by the body towards implantation produced fibrosis. PCL/Ag, considered a non-toxic material as it does not degrade tissues, has an inflammatory process like that of a common injury. However, the continuous presence of the implant in the tissue causes the inflammatory and repair process to be prolonged, producing a fibrous layer [28].

The foreign body reaction indicates that there was a rejection by the Wistar rats of PCL/Ag. The foreign body reactions may be caused by an excessive amount of implanted material, which, as previously discussed, was the case. At the time of implantation of the PCL/Ag, protein adsorption was carried out from different pathways activated by the initial injury. The main pathways involved to start the healing process are the coagulation pathway, the complement system, the fibrinolytic system and platelet aggregation [29]. Once the pathways are activated, the generation of a provisional matrix begins at the site of the implantation of the material, mainly made of fibrin. It is fibrin that begins the angiogenesis process to enhance healing [30]. Subsequently, an acute inflammation of short duration (hours/days) occurs, where leukocyte migration (mainly neutrophils) occurs from the blood vessels [25,26,27,28,29,30,31]. 

With the arrival of neutrophils at the implantation site, an attempt is made to phagocytose the PCL/Ag, a case that is impossible because the size of the implant is greater than the neutrophils, a process called “frustrated phagocytosis”. Since the event was not resolved, it gives way to chronic inflammation characterized by the presence of lymphocytes, monocytes, macrophages, plasma cells and giant cells. These cells cover the material, generating a fibrous layer with the intention of isolating the material from the organism as shown in Figure 7 [32]. It seems that, although it cannot be ensured that PCL/Ag is 100% biocompatible with the subcutaneous tissue of Wistar rats, it can be deduced that at least it does not cause harmful effects in tissues adjacent to the material once it is implanted.

## 4. Conclusions

PCL/Ag nanofibers were manufactured using the electrospinning technique with a nanometric size of 400 ± 100 nm. The diameter of the nanofibers decreased as silver was added to their conformation due to an increase in the conductivity of the solutions. Scanning electron microscopy showed that the nanofibers are smooth, cylindrical and randomly oriented. IR spectrum, Raman and EDX showed that the PCL composite is embedded with silver. The wounds made in the implantation area healed completely at four weeks. PCL/Ag nanofibers generated chronic inflammatory infiltrate in the subcutaneous tissues of Wistar rats at 2, 4 and 6 weeks after implantation. A foreign body reaction to the PCL/Ag nanofibers was generated by the Wistar rats. The composite was not biocompatible because it generated an inflammatory infiltrate in the subcutaneous tissue of the Wistar rats, accompanied by a foreign body reaction. In the same way, thanks to this study, it is recommended to carry out future studies by placing single-layer nanocomposite implants and in a smaller quantity. This is recommended so that the reaction to foreign bodies is not carried out.

## Figures and Tables

**Figure 1 polymers-13-01135-f001:**
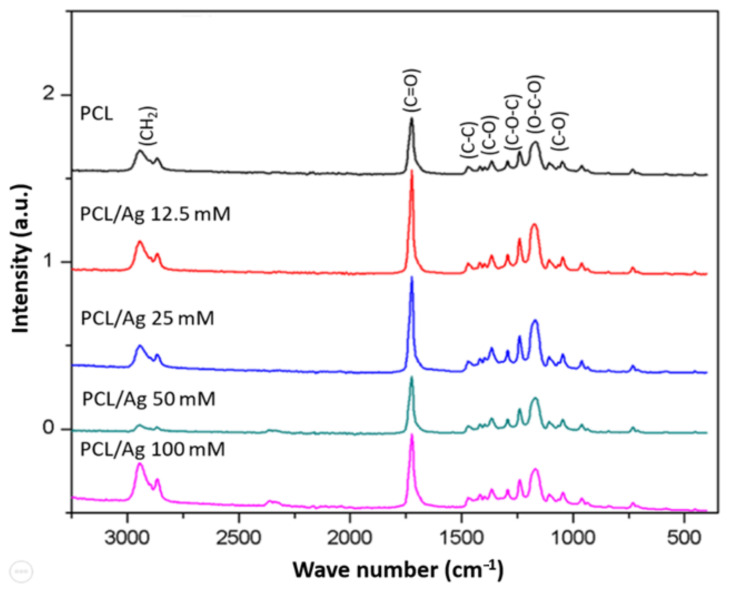
Infrared spectrum obtained from the different nanofibers of PCL/Ag (poly-ε-caprolactone/Ag).

**Figure 2 polymers-13-01135-f002:**
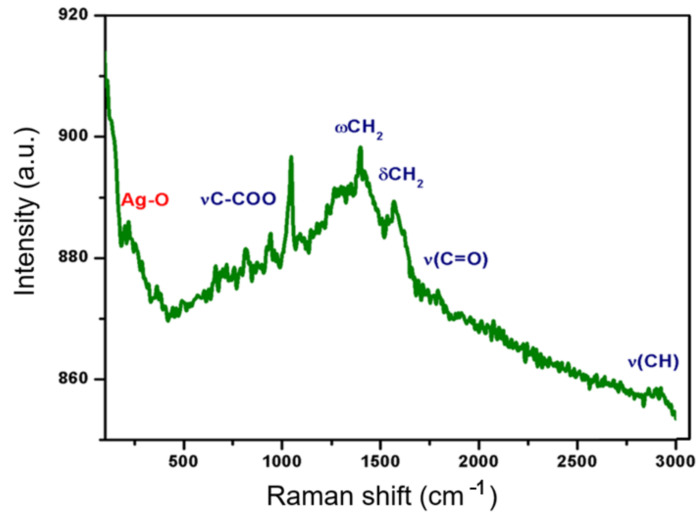
Raman spectrum obtained from PCL/Ag nanofibers.

**Figure 3 polymers-13-01135-f003:**
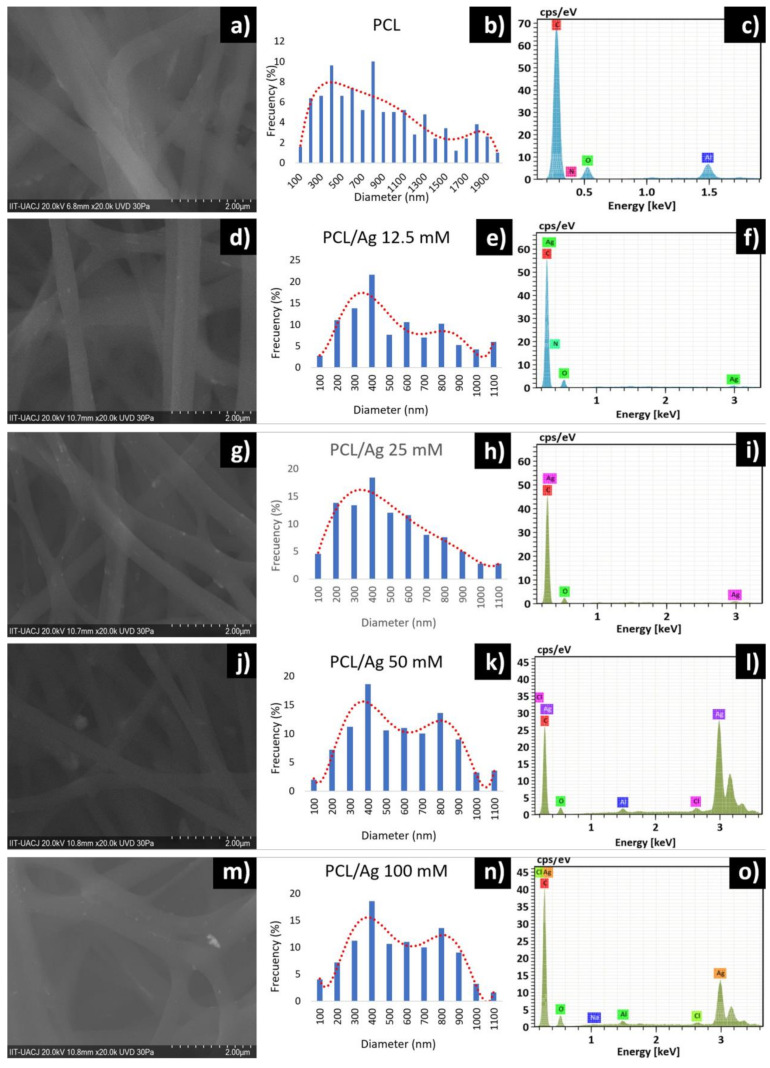
Micrographs obtained by SEM of the nanofibers: (**a**) PCL, (**b**) PCL frequency distribution, (**c**) PCL EDX spectrum, (**d**) PCL/Ag 12.5 mM, (**e**) PCL/Ag 12.5 mM frequency distribution, (**f**) PCL/Ag 12.5 mM EDX spectrum, (**g**) PCL/Ag 25 mM, (**h**) PCL/Ag 25 mM frequency distribution, (**i**) PCL/Ag 25 mM EDX spectrum, (**j**) PCL/Ag 50 mM, (**k**) PCL/Ag 50 mM frequency distribution, (**l**) PCL/Ag 50 mM EDX spectrum, (**m**) PCL/Ag 100 mM, (**n**) PCL/Ag 100 mM frequency distribution, and (**o**) PCL/Ag 100 mM EDX spectrum.

**Figure 4 polymers-13-01135-f004:**
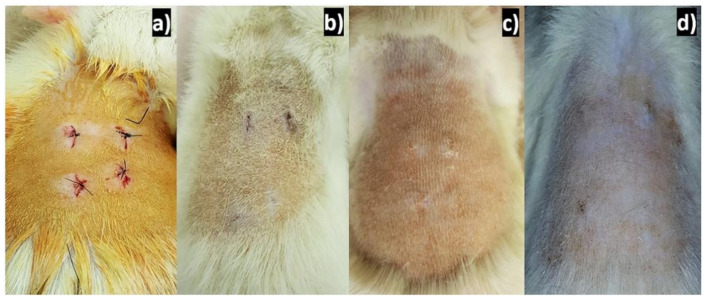
Surgical points in the dorsal area of Wistar rats after implantation: (**a**) just implanted the PCL/Ag, (**b**) two weeks after implantation, (**c**) four weeks after, and (**d**) 6 weeks after implantation.

**Figure 5 polymers-13-01135-f005:**
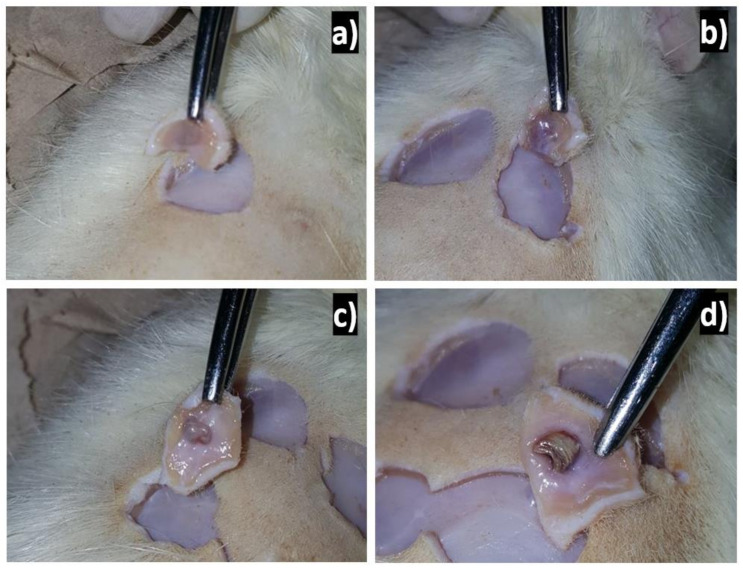
PCL/Ag nanofibers encapsulated in tissues of Wistar rats by a foreign body reaction 6 weeks after implantation: (**a**) 12.5 mM, (**b**) 25 mM, (**c**) 50 mM, and (**d**) 100 mM.

**Figure 6 polymers-13-01135-f006:**
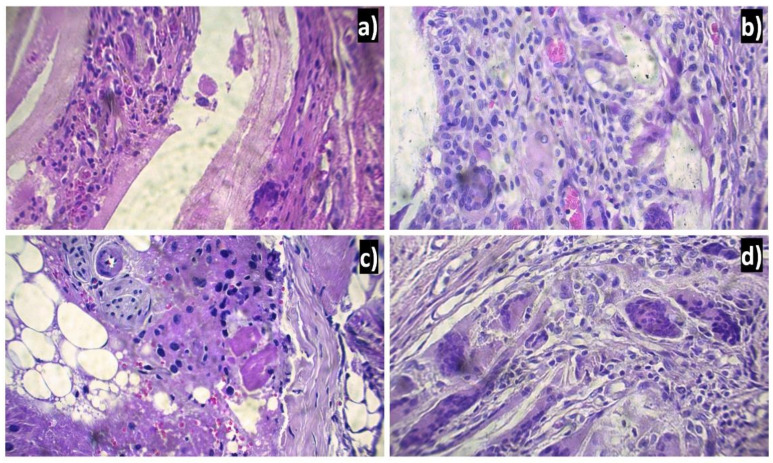
Histological preparation in tissue hematoxylin-eosin with the PCL/Ag seen at 40×: (**a**) 12 mM, (**b**) 25 mM, (**c**) 50 mM, and (**d**) 100 mM.

**Figure 7 polymers-13-01135-f007:**
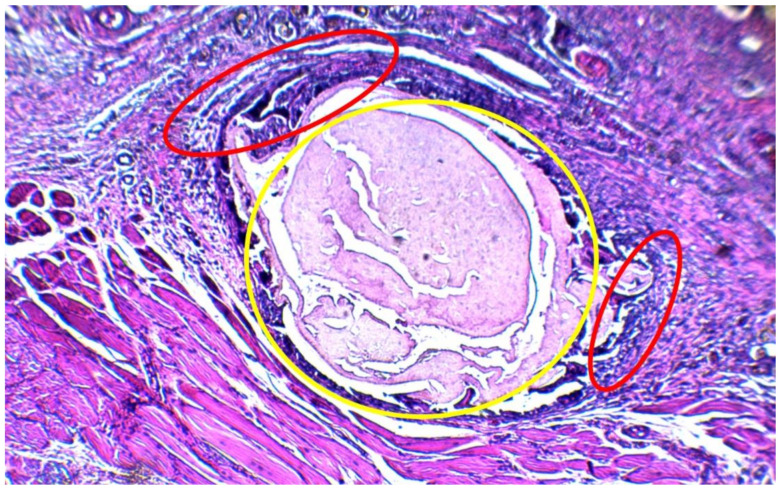
Foreign body reaction can be seen an organization of the different cells that make up the granuloma of the foreign body reaction. The nanocomposite of PCL/Ag is shown in yellow, and chronic inflammation with foreign body reaction is observed adjacent to it in red.

## Data Availability

The data presented in this study are available on request from the corresponding author.

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
