# Peer review of "Preliminary Biocompatibility Tests of Poly-ε-Caprolactone/Silver Nanofibers in Wistar Rats"

_polymers, 2021, doi:10.3390/polym13071135_

Round 1

Reviewer 1 Report

The authors studied the possibility to obtain a biocompatible material with antibacterial properties. They prepared poly-ε-caprolactone nanofibers embedded with silver nanoparticles by electrospinning technique. The obtained nanofibers were characterized by FTIR, Raman, dark field optical microscopy and SEM-EDX. Also, they performed biocompatibility and histological tests.

Considering that (i) the paper deals with an extensive studied polymer (poly-ε-caprolactone) and Ag nanoparticles and also, (ii) the manuscript conclude that the prepared "composite was not biocompatible by generating an inflammatory infiltrate in the subcutaneous tissue of the Wistar rats, accompanied by the foreign body reaction", this paper does not give any significant innovation and can’t be recommended for publication in Polymers.

For a future resubmission, some revisions are necessary:

1. In the whole text, the name of bacteria should be written with italic fonts.

2. Pages 4-5, lines 197-216: What is the relevance of FTIR analyses, since the presence of Ag is not put in evidence in any way?

3. For better view, I suggest to the authors to move Figure 2 between lines 225 and 226.

4. At page 6, line 242: Please add keV "...in the range of 1.5 keV belonging to aluminium...".

5. All references should be written between square brackets. I suggest to the authors to correct the references written else, in the whole manuscript (e.g. Johnston (2010), Atala (2010), Bhol and Shechter (2007), etc.).

6. At page 10, line 339: It is Figure 7, not 6. Please correct it.

Author Response

  1. In the whole text, the name of bacteria should be written with italic fonts.

Response: Text has been corrected and edited.

  1. Pages 4-5, lines 197-216: What is the relevance of FTIR analyses, since the presence of Ag is not put in evidence in any way?

Response: Text has been corrected and edited. New lines is added

In the IR spectrum of PCL-Ag samples no differences are observed, thus indicating no chemical modification of the structure of PCL, therefore, a physical union in the PCL-Ag follows.

  1. For better view, I suggest to the authors to move Figure 2 between lines 225 and 226.

Response: Text has been corrected and edited.

  1. At page 6, line 242: Please add keV "...in the range of 1.5 keV belonging to aluminium...".

Response: Text has been corrected and edited.

  1. All references should be written between square brackets. I suggest to the authors to correct the references written else, in the whole manuscript (e.g. Johnston (2010), Atala (2010), Bhol and Shechter (2007), etc.).

Response: Text has been corrected and edited

  1. At page 10, line 339: It is Figure 7, not 6. Please correct it.

Response: Text has been corrected and edited

Reviewer 2 Report

The manuscript entitled: “Preliminary biocompatibility tests of poly-ɛ-caprolactone/silver Nanofibers in Wistar rats” is an interesting study about the biocompatibility of this kind of Nanofibers in rats. I recommend the publication of the manuscript after elucidate some details:

- In the physicochemical characterization of the Nanofibers I recommend the authors to complete this characterization with H-NMR spectroscopy, in order to elucidate the functional groups of each molecule involved in the nanosystem formulation;

- I recommend the authors to determine de zeta potential of the Nanofibers by Electrophoretic Light Scattering, in order to evaluate the stability of the nanossystems;

- Before to perform in vivo studies, it would be great to perform in vitro cytotoxicity studies in fibroblasts;

- What kind of therapeutic application the authors intend with these Nanofibers?

Author Response

- In the physicochemical characterization of the Nanofibers I recommend the authors to complete this characterization with H-NMR spectroscopy, in order to elucidate the functional groups of each molecule involved in the nanosystem formulation;

Response: Further characterization by NMR reveals new functional groups, because we are not making changes in the structure of the polymer, is only a particulate mixture and PCL

 - I recommend the authors to determine de zeta potential of the Nanofibers by Electrophoretic Light Scattering, in order to evaluate the stability of the nanossystems;

Response: Text has been corrected and edited. New lines are added

The silver nanoparticles were synthesized by this simple reduction method give a slightly viscous reddish-brown solution was obtained which, when diluted, turned yellow. the particle size was determined by dynamic light scattering is around 6 ± 2.9 nm with zeta potential of -55 ± 3.4 mV, indicating that they have good stability.

- Before to perform in vivo studies, it would be great to perform in vitro cytotoxicity studies in fibroblasts;

Response:  

This paper is a continuation of the type of materials that we have synthesized and investigated, additional information can be found in the following publication.

Monrreal-Rodríguez, A. K., Garibay-Alvarado, J. A., Vargas-Requena, C. L., & Reyes-López, S. Y. (2020). In vitro evaluation of poly-ε-caprolactone-hydroxypatite-alumina electrospun fibers on the fibroblast’s proliferation. Results in Materials, 6, 100091.

- What kind of therapeutic application the authors intend with these Nanofibers?

Response:  

New composites with improved bactericidal properties used for dressings for skin and tissue regeneration

Reviewer 3 Report

The MS by Álvarez-Ortega demonstrates the synthesis PCL-Ag nanofibers (400 ± 100 nm)  using the electrospinning technique. The nanofibers are well characterized by SEM, IR, Raman spectroscopies and were placed in the subcutaneous tissue of the rats. The rats showed good healing, with the lesions completely healed at 4 weeks after implantation. Overall, this is a nice piece of work and the experiments are well-designed and conducted. The data supports the claims. I recommend publication in the current form.

Author Response

Response: Text has been corrected and edited 

Round 2

Reviewer 1 Report

The authors have made the suggested small corrections. Overall, the manuscript is interesting and well presented, but, in my opinion, is not suitable for publication in Polymers. I suggest to the authors to submit the manuscript to other journal, oriented to medical/analytical applications.

Reviewer 2 Report

I recommend the publication of the mnauscript in the present form.